# Adapting Deep-Learning Audio Models for Abnormal EEG Classification

Yixuan Zhu
*School of Engineering*
*University of the West of England*
Bristol, UK
yixuan2.zhu@live.uwe.ac.uk

David Western
*School of Engineering*
*University of the West of England*
Bristol, UK
david.western@uwe.ac.uk

*Abstract*—EEG signal analysis and audio processing, though distinct in application, share inherent structural similarities in their data patterns. Recognizing this parallel, our study pioneers the application of two renowned audio processing models, PaSST and LEAF, to the realm of EEG signal classification.

In our experiments, the adapted PaSST and LEAF models delivered exceptional performance on the Temple University Hospital Abnormal EEG Corpus (TUAB). Specifically, PaSST achieved an impressive accuracy of 95.7%, while LEAF registered 94.0%, both substantially outstripping previously established benchmarks. Such achievements underscore the potential of tapping into cross-domain models, particularly from the audio sector, for advancing EEG research.

Notably, while these larger audio models brought about unparalleled results, maximizing their capabilities required addressing the limitations of available EEG data volume. Thus, we introduced innovative pre-training strategies derived from diverse datasets, further enhancing the performance efficacy. With these refinements, PaSST reached a landmark accuracy of 96.1% on the TUAB dataset, marking a significant stride forward in EEG signal processing.

By leveraging the intrinsic resemblances between EEG and audio signals, we have successfully repurposed these audio models. We recommend further work devoted to the exploration of the transferability of machine learning audio techniques to healthcare time series tasks.

*Index Terms*—Machine Learning for Healthcare, EEG Classification, Deep Learning, Audio Models

## I. INTRODUCTION

Historically, models designed for EEG analysis have predominantly been lightweight, a reflection of both computational considerations and the prevailing notion that EEG data, being seemingly less intricate than images or audio, can be decoded effectively with simpler architectures. However, this perspective might be inherently limiting. In reality, EEG signals, though different in nature from images or audio, possess rich information content. Hence, compact models may struggle to learn intricate and global mappings crucial for deciphering EEG patterns, especially when considered in relation to the evolving nature of neural network architectures. Over the years, models like transformers have showcased the prowess of larger and more computationally intensive architectures in learning global mappings and relationships within the data. These architectures, originally tailored for domains like natural language processing (NLP), have found their relevance in diverse applications, including audio processing.

The advancements in audio classification have led to intriguing possibilities in EEG analysis. By integrating robust and comprehensive modeling techniques from audio classification, we can address some limitations of existing lightweight EEG-specific models. This integration may lead to meaningful improvements in EEG analysis. To this end, we explored two pioneering audio classification models: PaSST [11] and LEAF [26]. Notably, while LEAF is conventionally a front-end, it is conventionally integrated with EfficientNet to achieve the versatility of a holistic model. After essential modifications and fine-tuning, these models exhibit state-of-the-art performance on the TUAB dataset, solidifying our hypothesis.

Although the availability of clinical EEG data is relatively limited, compared with audio data, we addressed this challenge through two pretraining strategies. First, we enhanced our dataset by using label extraction from EEG reports through NLP techniques, as detailed in previous work on auto-labelling [24]. This approach provided us with an additional labelled dataset, leveraging our established methodologies. Second, we applied cross-domain pretraining on speech datasets. The rationale behind this was that the models might learn relevant temporal characteristics of one-dimensional time series from the speech data. These approaches were beneficial for our models due to their higher capacity, enabling them to better learn and adapt to EEG tasks. The combination of these strategies not only supplemented our training data but also ensured that our models fully utilized their advanced capabilities in EEG analysis.

In summary, this research signifies a notable advancement in the field of EEG data processing. It highlights the potential for improved diagnostic accuracy for neurological disorders through the use of advanced, and often more computationally demanding, architectures.

*Generalizable Insights about Machine Learning in the Context of Healthcare*

This study confirms that large models from domains such as audio processing, which are relatively well-resourced in terms of data and development effort, can be effectively applied to relatively data-scarce time series fields like abnormal

EEG classification, leveraging the extensive development work and advanced architectural designs from the better resourced domains. We also explore the use of pre-training with both same-domain and cross-domain (audio pre-training for EEG) datasets, demonstrating benefit in the same-domain case only. Our results are particularly relevant to other healthcare time series fields, such as ECG or ECoG, which face similar challenges of data scarcity and low signal-to-noise ratios.

Additionally, our work disproves a widely held assumption that the performance ceiling for our dataset (Temple University Hospital Abnormal EEG Corpus) would match typical clinical inter-rater agreement levels. We provide evidence that this performance ceiling does not apply when labelling is conducted by consensus among a panel. The same point will likely be important in assessing the performance ceiling of other clinical datasets, where the labelling practice applied to the research dataset does not completely mimic conventional clinical practice.

Although our attempt to enhance EEG data learning with cross-domain audio pre-training did not succeed as anticipated, this does not entirely discount the effectiveness of the principle that concepts embedded in one time series domain may be transferrable to another. This negative result may be due to too many training epochs, which could have overshadowed the benefits of pre-training on audio data. This lesson reminds us to adjust our training strategies and number of epochs more carefully when using cross-domain pre-training methods. Furthermore, it may be that, for cross-domain time series pre-training to work effectively, the pre-training dataset must cross multiple domains in order to achieve good generalization. As the importance of large foundational models grows in many machine learning applications, we consider the possibility

of developing foundational models for time-series analysis, trained on a very large range of time series data and fine-tunable to domain-specific tasks (just as many foundational language models are now being fine-tuned for more specific language tasks). We plan to continue exploring this area in future research to fully utilize the potential advantages of pre-training.

## II. Related Work

### A. Previous Models for EEG Data

While EEG analysis approaches, predominantly based on CNNs [3,7,15,17,22], have shown some success, they often do not fully capitalize on the advanced techniques widely used in other time-series domains. These domains have seen the adoption of various innovative models [11,21,26], going beyond traditional CNN frameworks, which have yet to be extensively applied to EEG data, especially within the TUAB dataset. The TUAB dataset, with its extensive and diverse EEG recordings, presents an ideal opportunity for the integration of such cutting-edge methodologies. However, the application of these advanced techniques in the context of TUAB has been notably limited, indicating a significant gap in leveraging state-of-the-art time-series analysis methods in EEG research.

Some recent studies have explored how to improve traditional EEG models. Among these studies, the work of [10] is particularly noteworthy. They demonstrated that merely increasing the depth (adding additional convolutional layers) or width (increasing the number of filters) of the models does not significantly enhance the performance of EEG models. This finding suggests the limitations of traditional methods in improving EEG analysis. Although there are examples of transformer applications in the field of EEG analysis [20,19,12], their use is not yet mainstream. Moreover, their potential for detecting abnormal EEG patterns has not been fully explored or applied.

### B. Cross-Domain Applications of Audio Techniques in EEG Research

In prior research, techniques initially designed for audio processing have been effectively repurposed to facilitate EEG analysis, primarily due to the shared temporal characteristics of the data types in both domains. For the same reasons, machine learning architectures that have proven to be successful in audio applications have also been successful in EEG, although we are not aware of any previous EEG studies translating machine learning models designed specifically for audio.

From a preprocessing perspective, given EEG data's temporal signal properties, several audio processing techniques have been adopted. These techniques range from windowing procedures to spectral decompositions, such as Fast Fourier Transform (FFT)[4] and Mel-frequency cepstral coefficients (MFCC)[5]. Furthermore, techniques for time-frequency image extraction, such as Short-Time Fourier Transform (STFT)[18] and Wavelet Transforms [2], have been adopted.

TABLE I: Summary of state-of-the-art performance metrics for different models applied to abnormal EEG classification on the TUAB dataset. All results are compared at the recording level using the official test set provided by TUAB. The newly proposed models and the highest score in each column are listed in bold. Among these models, 'Scope and Arbitration' (*) comprises multiple machine learning stages, whereas the others are single-stage models. This comprehensive comparison ensures that the evaluation is consistent and adheres strictly to the dataset's intended use for benchmarking.

| Model | Accuracy | Sensitivity | Specificity |
|---|---|---|---|
| 1D-CNN (T5-O1 channel) [25] | 79.3 % | 71.4 % | 86.0 % |
| 1D-CNN (F4-C4 channel) [25] | 74.4 % | 55.6 % | 90.7 % |
| Deep4 [17] | 85.4 % | 75.1 % | 94.1 % |
| TCN [7] | 86.2 % | | |
| ChronoNet [16] | 86.6 % | | |
| Alexnet [3] | 87.3 % | 78.6 % | 94.7 % |
| VGG-16 [3] | 86.6 % | 77.8 % | 94.0 % |
| Fusion Alexnet [1] | 89.1 % | 80.2 % | 96.7 % |
| Fusion CNN [15] | 89.8 % | 81.3 % | 96.9 % |
| Scope and Arbitration* [27] | 93.3 % | 92.0 % | 92.9 % |
| **LEAF-EEG** | 94.0 % | 92.4 % | 97.2 % |
| **PaSST-EEG** | 95.7 % | 94.8 % | **97.6 %** |
| **PaSST-EEG-Pretrained** | **96.1 %** | **95.5 %** | 97.5 % |

There are also methods specifically designed for feature extraction in the audio domain, like Linear Frequency Cepstral Coefficients (LFCC) [9], that have found use in EEG analysis.

On the model front, both EEG and audio research share an affinity for convolutional neural networks (CNNs)[3,7,15,17,22] and Transformer-based models [11,26], predominantly due to the temporal nature of the data involved. However, a specific model architecture used in the audio domain cannot ordinarily be translated to the EEG domain without any modifications. This could be attributed to the additional channel dimension present in EEG data, compared with 'mono' audio. Moreover, EEG data often pose interpretability challenges and arguably exhibit lower information density than their audio counterparts [14], which often necessitates the use of larger window sizes in EEG analysis. Coupled with the typically smaller data volumes in EEG research, these factors contribute to increased difficulty in training models within this domain.

### C. Original Models of PaSST and LEAF

In the audio domain, including tasks such as speech recognition and audio classification, high-performing model structures are largely based on EfficientNet and Transformers. Considering the model size and computational complexity, we chose two models that perform close to the state-of-the-art in the audio domain, PaSST and LEAF, as our base models.

As can be seen in Table II, the selected models represent a significant step up in complexity from traditional EEG models such as Deep4. These choices allow us to explore the potential of these more complex architectures in EEG signal classification tasks without overly taxing computational resources and data availability, and the work of Kiessner et al. demonstrates that a larger model does not necessarily mean better performance [10].

The PaSST (Parallel Attention Scoring and Selective Transformation) model takes time-frequency images as inputs [11]. It first applies the STFT (Short-Time Fourier Transform) to the one-dimensional time series, then processes it through a Mel filter, simulating human sensitivity to different audio frequency bands to form a more representative spectrogram. This method helps to highlight important features in speech signals. The main body structure of PaSST is based on Transformer. It uses a parallel attention mechanism to perform feature selection and transformation on the spectrogram, effectively extracts key information, and classifies it through a deep network.

On the other hand, the LEAF (A Learnable Frontend for Audio Classification) [26] frontend also takes time-frequency images as inputs. Before applying Gabor filters for spectrum processing, the audio signals undergo a Gaussian low-pass filtering followed by sPCEN (short-time Per-channel Energy Normalization), a technique commonly employed for dynamic range compression and background noise suppression, enhancing the salient features of audio signals while mitigating the effects of non-essential components [23]. Gabor filters, inspired by the cochlear model in natural signal processing, can simulate the perception and processing of audio signals by the human ear to obtain high-quality audio feature representations [6]. The main body structure in the original implementation of LEAF is based on EfficientNet, using depthwise separable convolution, bounded ReLU activation functions, and Squeeze-and-Excitation modules to achieve efficient feature extraction and classification.

The design principles and efficient performance of these two models make them good candidates for our experiments on EEG data. In the following sections, we will detail how we modified these two models to adapt to the characteristics of EEG signals and achieved significant performance improvements on the TUAB dataset.

## III. METHODS

### A. Data

*1) TUAB:* The experiment design and evaluation methods of this article are predicated on the TUAB dataset [13], which contains 2,993 EEG recordings labelled as normal/abnormal. These recordings are sourced from over 30,000 clinical EEG recordings collected by the Temple University Hospital from 2002 to the present, and all recordings use the standard 10-20 electrode placement system.

The TUAB dataset has been divided into training and evaluation sets, following an officially provided split. The training set contains 1,371 recordings labelled as normal' and 1,346 recordings labelled as abnormal'. The test set contains 150 normal recordings and 126 abnormal recordings. This structured division ensures a consistent framework for assessing and comparing model performances across studies.

In our experiments, we adopted a data preprocessing strategy similar to that of Deep4 [17]. Specifically, we used a 20-minute segment from the second minute to the 21st minute of each recording. When the length of a recording is less than 21 minutes, we extract from the second minute to the end of that recording. Then, we resampled the signals at 100 Hz and divided the recording into non-overlapping windows of one-minute duration. Therefore, after the windowing process, each recording generated 15-20 windows. As shown in Figure 2, the dimensions of each window are 21x6000.

Additionally, we selected 21 standard 10-20 channels for data collection and restricted the data range to be between -800 and 800 to remove obvious artefacts. This limitation was implemented using a clipping method, consistent with the approaches described in [17,7]. We did not use any other preprocessing methods, such as bandpass filters or data normalization. All of these ensure the originality of the data,

TABLE II: Comparison of Model Complexity and Computational Requirements. Total size = Input size + Forward/backward pass size + Params size

|  | Trainable params | Total size (MB) |
|---|---|---|
| Deep4 [17] | 303,452 | 32.86 |
| **LEAF-EEG** | 4,015,294 | 230.36 |
| **PaSST-EEG** | 89,188,610 | 1074.51 |

which is beneficial for us to evaluate the performance of the model on original EEG signals.

*2) AutoTUAB:* AutoTUAB, derived from the parent dataset TUEG, serves as an expanded dataset compared to TUAB. This was achieved using automatic labelling based on natural language processing of text reports, as described by [24]. We refer to this automatically labelled alternative to TUAB as 'AutoTUAB'. We selected only data with label confidence exceeding 99%, length greater than 6 minutes, and all the 21 desired channels. Following the screening process, AutoTUAB contained 26,504 recordings - 19,109 abnormal recordings and 7,395 normal recordings, across 18,747 sessions. Like in [27], to accommodate the unbalanced nature of the data we adopted a weighted loss function in the training of machine learning models. Cross-entropy was used as the loss function, and the contribution of each sample was given a weighting that was inversely proportional to the number of samples in its class. Compared to TUAB, the AutoTUAB dataset is larger and more diverse. It is arguably more representative of clinical data since it is not manually selected, whereas the examples in TUAB were selected to form a dataset conducive to machine learning [13].

In our research, we employ the AutoTUAB dataset for the pre-training of our model, ensuring that all samples overlapping with the TUAB dataset have been removed to prevent data leakage. Given that both AutoTUAB and TUAB are EEG datasets and the fact that we have designed identical training tasks, we postulate that AutoTUAB can assist the model in learning the essence of EEG signals. Further, it may even enable the model to discern between normal and abnormal EEG signals. This pre-training step with AutoTUAB, having excluded any duplicative samples, we believe, can help initialize the model in a more advantageous region, thus facilitating more effective subsequent supervised learning.

*3) AudioSet:* AudioSet [8], developed by Google's Sound Understanding group, is an extensive, publicly available dataset of over 2 million manually annotated 10-second audio clips drawn from YouTube videos. These clips span 632 unique audio event classes, including human and animal sounds, music, noise, and environmental sounds, making it a rich and diverse resource for audio-related machine learning tasks. The annotations are created through a rigorous manual process, ensuring their reliability. In our research, we employ AudioSet during the pre-training phase to help our model learn diverse audio features, thereby enhancing its ability to understand the specific characteristics inherent in EEG data. It should be noted that the use of AudioSet should comply with YouTube's Terms of Service.

### B. Modifying the PaSST Model

For the Transformer-based PaSST model (Figure 1a), we implemented several modifications to better align with the characteristics of EEG signals. First, we adjusted the parameters of the Short-Time Fourier Transform (STFT), including the sampling rate, window size, and window stride, ensuring

that its output dimensions meet the input requirements of subsequent modules. These adjustments allow the model to more effectively process EEG input. As illustrated in Figure 2, the input is initially split along the channel dimension, with STFT applied to each channel individually. The resulting data is then concatenated along the channel dimension. After applying STFT, the data dimensions are 513x286x2. By squaring and summing, the dimensions of the real and imaginary parts are reduced, resulting in final data dimensions of 513x286.

Second, we removed the Mel filter from the original PaSST model. While the Mel filter is commonly used in audio processing to simulate human auditory sensitivity to different frequencies, there is no evidence to suggest that the frequency distribution in EEG signals follows a similar pattern [9]. Therefore, we opted to exclude this component.

In place of the Mel filter, we added a two-dimensional adaptive pooling layer to the model to interface with subsequent classification modules. This adaptive pooling compresses the last two dimensions of the data to 128x282. There are two primary reasons for this modification. First, without the Mel layer, the frequency dimension of the STFT output does not align with the requirements of subsequent modules. Second, in the original PaSST, the time dimension was represented by $16,000 \times 0.025 = 400$ data points, whereas our current setup, without additional windowing, results in $60 \times 100 = 6,000$ data points. This substantial difference in the time dimension makes it challenging to integrate the modules seamlessly, even with changes to the STFT's stride and window size. Consequently, this pooling layer is necessary to compress and adjust the time dimension. Additionally, the pooling layer helps to transform high-dimensional feature maps into low-dimensional feature vectors, which aids in reducing overfitting.

Finally, we modified the first convolutional layer of the PatchEmbedding module to expand its input channels to 21. Patch embedding in Transformers involves splitting an image into fixed-size patches and converting each patch into a high-dimensional vector for sequential processing. With this adjustment, the convolutional layer is not only capable of extracting features from the time dimension but also fuses data across the 21 EEG channels. This design enables the model to capture intricate patterns across the EEG channels more effectively.

### C. Modifying the LEAF Model

For the EfficientNet-based LEAF model (Figure 1b), we similarly made some adjustments to make it better adapted to EEG signals.

First, we modified the parameters of the Gabor filter and Gaussian low-pass filter in the LEAF model, including window size and window stride. As shown in Figure 2, similar to STFT, the Gabor filter expands the time dimension of the EEG signal from one dimension to two dimensions, but it doesn't compress the time dimension (The stride is 1). After undergoing STFT, the input dimensions changed from 21x6000 to 21x80x6000. This modification allows the model to retain more information while meeting the input dimension

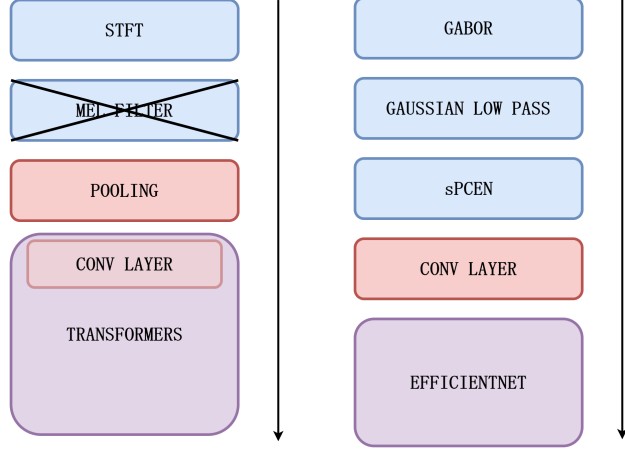

(a) Modified PaSST model.    (b) Modified LEAF model.

Fig. 1: Modified (a) PaSST model and (b) LEAF model. The blue and purple blocks represent the original components of the model, with blue indicating the preprocessing stage and purple indicating the feature extraction and classification stage. Crossed-out blocks signify components that were removed during the model's adaptation process. Newly introduced or modified components are represented by red blocks..

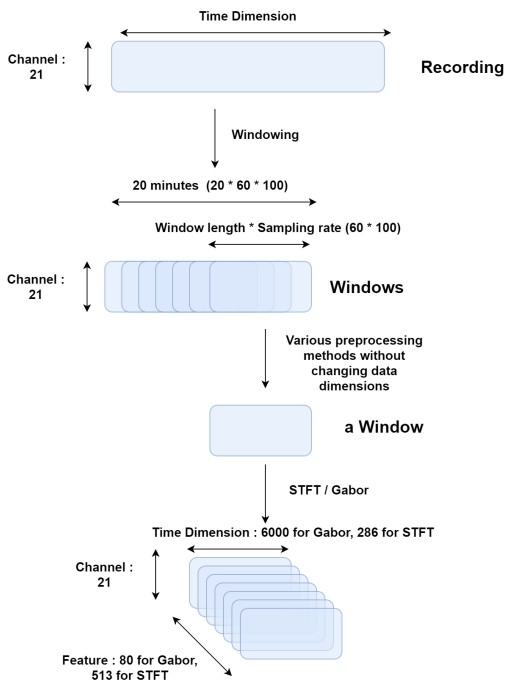

Fig. 2: Data input methods for LEAF and PaSST models. For the sake of clarity, the batch size dimension is not shown in the figure.

requirements of subsequent modules. This change is based on the characteristics of EEG signals, that is, the information distribution in EEG signals may differ from the audio signals in the original LEAF model. Then, as with PaSST, we modified the one-dimensional convolutional layer at the beginning of the classification module, expanding its input channels to 21.

Through the above adjustments, we successfully migrated the PaSST and LEAF models from the audio processing field to the EEG signal classification task. The next section will detail our experimental results on the TUAB dataset.

*D. Pre-training on the unlabelled dataset with NLP generated labels and cross-domain datasets*

Our study is motivated by the challenges inherent in applying advanced machine learning models to EEG signal classification, especially when faced with limited datasets. As shown in II, both PaSST-EEG and LEAF-EEG have parameter sizes that, while modest in comparison to many models in the Computer Vision (CV) or Speech domains, are still quite sizable for the TUAB dataset, which contains fewer than 3,000 recordings. We surmise that granting these models access to more data or providing them with a better initialization could enhance their training outcomes. This insight led us to propose two pre-training strategies: one on the larger and diverse AudioSet and the other on the AutoTUAB dataset. With these considerations in mind, we crafted a three-part experiment to gauge the merits of various training methodologies for EEG signal classification.

No Pre-training Group: Directly train and test on the TUAB dataset. This serves as our foundational comparison for assess-

ing the merits of enhanced training strategies.

AudioSet Pre-training Group: We conducted supervised pretraining on the AudioSet dataset, which offers a vast collection of well-defined labels for audio events. This pretraining provided our models with an initial understanding of temporal features, which can be highly beneficial for EEG classification when subsequently trained and tested on the TUAB dataset.

AutoTUAB Pre-training Group: Supervised pretraining was also performed on the AutoTUAB dataset, which originates from TUEG. Unlike TUAB, where labels are meticulously crafted through the combined evaluations of multiple experts, the labels in AutoTUAB were generated using an NLP model from clinical EEG reports. These NLP-generated labels are considered to be pseudo-labels, as they are not directly annotated by human experts but are inferred by the model. These pseudo-labels can be speculative and potentially biased, reflecting the inherent challenges in working with clinical data. To mitigate these issues, we introduced a filtering mechanism, retaining only those recordings where the probability of abnormality labels is either below 0.01 or above 0.99, thereby increasing our confidence in the labels used. Despite their limitations, we believe these pseudo-labels can still serve as valuable proxies, helping the model familiarize itself with the general characteristics of EEG signals and effectively distinguish between abnormal and normal patterns.

In our research, we primarily utilized the PaSST model. While both PaSST and LEAF represent the broad spectrum of cutting-edge temporal signal processing methods, PaSST

was chosen due to its superior performance in the absence of pretraining.

To thoroughly assess the impact of various pre-training datasets, each model was tested under all three experimental conditions, with each condition being reiterated five times to neutralize any potential discrepancies from random variations. All pre-trained models were fine-tuned on the entire network during the formal training phase, which consisted of 30 epochs, but incorporated early stopping to ensure that any benefits derived from pre-training were not negated by overfitting to the TUAB training set. This stringent testing procedure offers a comprehensive insight into the effects of different pre-training strategies across a range of temporal signal processing paradigms.

Furthermore, it's essential to highlight that the benefits of pre-training can occasionally be overshadowed when there's an abundance of primary training data. As such, we conducted supplementary experiments to examine the potency of our two pre-training methods when faced with limited training data availability. To achieve this, we evaluated the performance of PaSST models using scaled-down portions of the training dataset, specifically at 1, 0.8, 0.6, 0.4, and 0.2 of its original size. To minimize the impact of random fluctuations, each of these configurations was replicated five times. Taking into account the three pre-training strategies we deployed, including the baseline method, we executed a grand total of 75 experimental trials. This rigorous approach further elucidates the significance and potential advantages of our proposed pre-training techniques, especially in scenarios where training data might be scarce.

### E. Experiments

For each model, we conducted five independent experiments on the TUAB dataset's training and test sets, with each experiment using a different random seed to ensure the credibility of the results. Ultimately, we took the average result from these five experiments on the test set as the final performance of each model. Furthermore, we conducted a hyperparameter search for both models, testing three different learning rates (LR): 0.001, 0.0001, and 0.00001. All evaluations on the test set were performed at the recording level. We employed a simple mean to aggregate the results from the window level to the recording level.

## IV. RESULTS

As shown in Table I and III, the adapted PaSST model achieved an average accuracy rate of 95.7% (The standard deviation is 0.0055) on the EEG signal classification task when using an LR of 0.00001, while the adapted LEAF model reached an average accuracy of 94.0% when using an LR of 0.0001. Compared to the best result of 89.8% achieved by previous one-stage models on this task, our models showed improvements of 5.9% and 4.2%, significantly surpassing previous research.

These results demonstrate the effectiveness of the modified PaSST and LEAF models in EEG signal classification. This

further validates our hypothesis: audio classification models, with appropriate adjustments, can be effectively applied to EEG signal classification tasks.

Moreover, as shown in Figure 3, the PaSST model under three different pre-training conditions — 'No pretraining', 'Pretraining on AudioSet', and 'Pretraining on AutoTUAB' — achieved accuracies of 95.7%, 95.9%, and 96.1% (The standard deviation is 0.0076) respectively on the TUAB test set after formal training with the complete TUAB training dataset. As the proportion of the TUAB training set used decreased, the model's accuracy for all three scenarios gradually declined. However, the gap between Pretraining on AutoTUAB and the other two methods progressively widened. Ultimately, when utilizing only 0.2 times the TUAB training set, the PaSST model's final performances under the three pre-training methods were 80.1%, 80.1%, and 82.3% respectively.

We conducted our experiments using an NVIDIA RTX 4090 GPU. The training and testing times for each model are summarized in Table IV.

## V. DISCUSSION AND FUTURE WORK

### A. Adaptation of Time Series Models for EEG Analysis

Our research successfully demonstrates the potential of applying and adapting audio processing models, specifically PaSST and LEAF, for EEG signal classification tasks. This achievement not only illustrates the flexibility and adaptability of these audio models but also suggests the existence of common characteristics shared between audio and EEG signals. Going forward, we plan to experiment with more models or modules from various time series domains for application in the EEG field. Through this approach, we aim to further investigate the interplay and shared features among different types of time series data and EEG signals, ultimately

TABLE III: The effect of learning rate on the performance (Accuracy) of PaSST and LEAF on TUAB, compared with Deep4 as a baseline model. No pre-training is used in these cases.

| Model | Learning Rate | | |
|---|---|---|---|
| | 0.001 | 0.0001 | 0.00001 |
| Deep4 | 0.854 | 0.860 | 0.800 |
| EEG-PaSST | 0.837 | 0.949 | 0.957 |
| EEG-LEAF | 0.880 | 0.940 | 0.878 |

TABLE IV: The table below provides a comparison of the training and testing times for our adapted models (PaSST and LEAF) and a baseline lightweight model (Deep4). Training Time: The average time taken to train each epoch on the training set, which consists of 2717 samples. Each model was trained for 30 epochs. Test Time: The time taken to run inference once on the test set, which contains 276 samples.

| | Epoch Training Time (s) | Test Time (s) |
|---|---|---|
| Deep4 | 942 | 4.3 |
| EEG-PaSST | 639.6 | 102.5 |
| EEG-LEAF | 2099.9 | 32 |

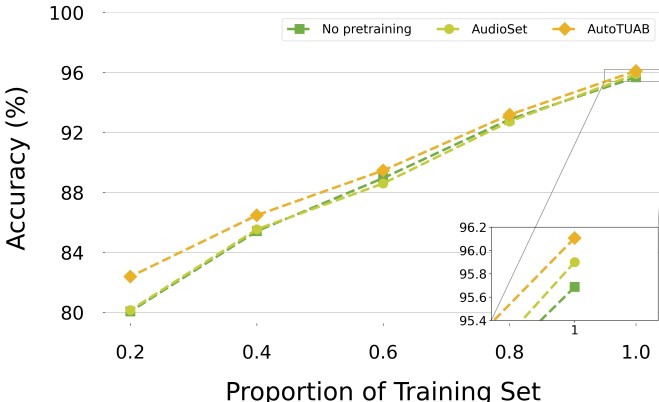

Fig. 3: We subjected PaSST to pre-training on both the Auto-TUAB and AudioSet datasets. Subsequently, formal training was executed on varying proportions of the TUAB training set, and the final evaluation was performed on the TUAB test set. The figure displays the variation in model performance (Accuracy) as a function of the usage ratio of the TUAB training dataset. Learning Rate = 0.00001.

enhancing the models' understanding and interpretation of EEG signals.

### B. Assessing the Impact of Pretraining Sources on EEG Classification

As shown in Figure 3, we explored whether pretraining can enhance the performance of our adapted audio model for EEG classification. It's evident that pretraining on AudioSet didn't significantly benefit the model's training for the EEG classification task. On the other hand, pretraining on Auto-TUAB consistently improved the model's performance on the TUAB test set, especially when the amount of training data was limited. The efficacy of pretraining on AutoTUAB is understandable given that AutoTUAB and TUAB are of the same data type and tackle the same tasks, even though the labels in AutoTUAB might be biased.

We believe that the lacklustre results from AudioSet stem partly from the significant divergence in tasks – the frequency band in which human speech operates is quite distinct from the EEG data frequency band. Moreover, the classification of human speech may not necessarily require the model to deeply comprehend temporal concepts (which might explain why studies based on AudioSet usually employ small windows).

However, the underwhelming performance of AudioSet-based pretraining does not conclusively negate the potential of audio data to assist EEG tasks in learning temporal features. In future endeavours, we're considering the design of self-supervised tasks on AudioSet, thereby using it to initialize models with less emphasis on discriminating task-specific classes.

### C. Revisiting the Use of Larger Models in EEG Analysis

Our work reveals the potential for larger models to be employed effectively for EEG signal analysis tasks. Con-

ventional wisdom, influenced by the early success of Deep4 [17], has advocated for smaller models in the context of EEG signal processing, citing the challenges posed by the substantial input size and the limited availability of clinical EEG data [1]. This limited architectural scope has led to the perception of a performance ceiling [7,10] below 90% accuracy in clinical EEG classification. Our study presents evidence that with larger models, even without the benefit of pre-training, previously perceived performance limits can be exceeded. This significant advancement over the previous state-of-the-art models underscores the promise and relevance of our approach.

### D. Future Directions for Model Optimization

The improvements to the audio models in this study were straightforward, leaving room for further optimization. Currently, only one convolution layer is used to process inter-channel relationships, which may not fully capture complex interactions. In future work, we will explore advanced methods, such as introducing a self-attention mechanism, to better represent these relationships and enhance model performance.

While we adjusted parameters and removed modules in the feature extraction stage, these changes may not fully address the unique characteristics of EEG signals. We plan to introduce more effective feature extraction modules tailored for EEG processing to further improve model performance.

At the window level, we simply average the results across windows to obtain recording-level predictions. Zhu et al. [27] suggest using machine learning-based arbitration for combining results at the recording level, which we will explore in future work.

We also aim to simplify the model and enhance computational efficiency by employing techniques like model pruning, knowledge distillation, and optimized self-attention mechanisms to reduce computational complexity without sacrificing accuracy.

Lastly, testing on other datasets, such as the Harvard Electroencephalography Database [28] and the South Asian NMT dataset [29], will help evaluate the generalizability of our models. Extending the evaluation to different EEG datasets and other healthcare time-series data, like ECG and ECoG, will further validate the model's applicability across clinical scenarios.

### VI. Conclusion

In this study, we delved into the intricacies of adapting two high-performing audio processing models, PaSST and LEAF, for EEG signal classification tasks. The resulting performance improvements were both encouraging and substantial. Specifically, the PaSST model achieved an impressive accuracy of 95.7% on the TUAB dataset without any pretraining. However, when we further pre-trained PaSST on AutoTUAB, its performance was boosted to an even higher accuracy of 96.1%. In contrast, the LEAF model exhibited an accuracy of 94.0%. To put this in context, these results mark a notable progression

from previous benchmarks set by one-stage models, such as the Fusion CNN, which had an accuracy of 89.8%.

Our experimentation with pretraining strategies offered unique insights. While the use of AudioSet as a pretraining medium didn't yield a significant performance enhancement, the AutoTUAB dataset emerged as a powerful enabler, especially when dealing with limited training data. This accentuates the potential of domain-specific pretraining in delivering enhanced results.

This research provides further evidence that machine learning accuracy can exceed inter-rater agreement levels between human experts. Hence further effort should be devoted to the translation of such techniques into clinical practice to optimise the efficiency and quality of healthcare delivery.

### DATA AND CODE AVAILABILITY STATEMENT

The code for the experiments is available at https://github.com/zhuyixuan1997/Audio_model_for_EEG. The Temple University Hospital EEG Corpus (TUH EEG) dataset used for model training and evaluation can be accessed at https://isip.piconepress.com/projects/tuh_eeg/.

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
