# OpenReview forum: "Adapting Deep-Learning Audio Models for Abnormal EEG Classification"
_IEEE.org/EMBS/BHI/2024/Conference — IEEE BHI'24_

### Official Review · Reviewer_zspc · 2024-08-07
**Adapting Deep-Learning Audio Models for Abnormal EEG Classification**

**Overall Rating:** 6
**Confidence:** 4

**Other Quality Metrics:**

(a) Clarity of Writing: Good
(b) Clinical Significance: Fair
(c) Methodological Novelty: Good
(d) Experiments and Results: Good

**Questions For The Authors:**

1. Could you provide more details on the specific modifications made to the PaSST and LEAF models for adapting them to EEG data?
2. Can you discuss any potential biases or limitations in your dataset that might affect the generalizability of your results?

**Strengths:**

1. The adapted models, including PaSST, achieve state-of-the-art performance on the TUAB dataset, far exceeding existing benchmarks.
2. The introduction of pre-training strategies to enhance model performance despite limited EEG data demonstrates considerable potential.

**Summary Of The Paper:**

The paper explores the application of two audio processing models, PaSST and LEAF, to EEG signal classification.
This work evaluates the approach of adopting deep-learning audio models for abnormal EEG classification, highlighting strengths, identifying weaknesses, and posing questions to clarify understanding and improve future research directions.

**Weaknesses:**

1. Despite the innovative pre-training strategies, the reliance on relatively small EEG datasets remains a limitation, potentially affecting the generalizability of the findings.
2. The complexity and computational demands of the adapted models could limit their accessibility and practicality for some researchers and clinicians.

---

### Official Review · Reviewer_FNBu · 2024-08-10
**overall arguments/findings of the paper are interesting but can be made more compelling via transformer-based EEG benchmarking (to see benefit of audio domain inspiration) and external validation of results (to assess overfitting due to large model size).**

**Overall Rating:** 7
**Confidence:** 5

**Other Quality Metrics:**

- (a) Clarity of writing - fair
- (b) Clinical Significance - good
- (c) Methodological Novelty - fair
- (d) Experiments and Results - good

**Questions For The Authors:**

- Q: what exactly makes EEG less information dense than audio? Seems like a controversial statement that warrants slightly more context. (although current study does not make this claim and only refers to it)
- Q: How is the inter-rater agreement related performance ceiling disproven by evidence presented? Pretraining on large-scale data/noisier labels is likely to only reduce its impact but not disprove it.
- Q: what do "one-stage" models mean, and how are PaSST/LEAF different?

**Strengths:**

- borrowing and adapting relevant ideas/models/data from audio domain for building EEG encoders is a promising and interesting direction of exploration
- study provides evidence of "model parameter/size scaling" and reinforces benefits of self-supervision/pretraining for EEG signals domain in low-data regimes

**Summary Of The Paper:**

study tests the hypothesis that audio related models can improve EEG encoding due to semantically related data. Empirical evidence shows that PaSST/LEAF audio-based models improve normal vs abnormal EEG classification beyond the presumed 87% barrier, even without pretraining.

**Weaknesses:**

- writing suggestion: abstract can be revised to first state the unaddressed need in improving EEG encoders/TUAB performance, the inspiration/hypothesis of adapting audio domain components, followed by proposed approach, empirical results, and study significance.
- writing suggestion: introductory argument and related work is weak/hand-way and not rigorous enough to motivate the study. Authors should mention what exactly are the temporal characteristics of EEG that existing EEG models/features fail to capture but the audio models can capture well, which would then clearly insipire the central hypothesis (audio --> EEG transfer of ideas/data/models).
- if study wants to claim that using larger complex models from audio is beneficial, past large/complex transformer-based EEG models, although recent, must be added as baselines when presenting audio models-based evidence. I'm not fully convinced that there is any inspiration taken from the audio domain after all the modifications are done. Mel filters from PaSST were removed, meaning there is nothing about audio remaining in the modified PaSST architecture? Tweaking filter/pooling dimensions should not be considered a major methodological improvement/adaptation. It is likely that the apparent increases in performance are simply due to model scaling via the transformer architecture, which is why those baselines are needed.
- there is no detail given on how the pretraining was done i.e. the self-supervised loss function. this is arguably a very important methodological choice that will affect all pretraining and fine-tuning normal vs abnormal results. For example, see - https://proceedings.mlr.press/v158/wagh21a/wagh21a.pdf, https://iopscience.iop.org/article/10.1088/1741-2552/abca18
- perhaps a more approriate interpretation of the pretraining findings may be that large-scale pretraining and large-scale models can help EEG encoders reduce the presumed effect of inter-rater disagreement. I'm not sure if model performance and inter-rater agreement levels are directly comparable.
- EEG models are known to have robustness and overfitting issues due to subject-varying and hardware-varying signal amplitudes. The lack of external validation of these results makes the ~95% accuracy results less reliable/trustworthy. This validation is especially important since no data normalization was done as part of preprocessing. Perhaps a dataset like https://www.frontiersin.org/journals/neuroscience/articles/10.3389/fnins.2021.755817/full or https://bdsp.io/content/harvard-eeg-db/2.0/ could be useful.

---

### Official Review · Reviewer_QBgV · 2024-08-11
**Adapting Deep-Learning Audio Models for Abnormal EEG Classification**

**Overall Rating:** 7
**Confidence:** 4

**Other Quality Metrics:**

Clarity of writing: good
Clinical Significance : NA
Methodological Novelty : great
Experiments and Results : good

**Questions For The Authors:**

In the state of the art only convolutional inference models are evaluated, why ?
Why in future work, you will explore more methods to further enhance the performance of the model ?
The optimisation next step shouldn't it be the simplification of models in order to reduce classification time for the same accuracy ?

**Strengths:**

The adopted methodology to validate PaSST and LEAF inference models for EEG signals.
The assessing of the impact of pretraining data on EEG classification,

**Summary Of The Paper:**

The authors present the ability to transfer audio classification models like PaSST and LEAF to the EEG signal classification. The database used is the Temple University Hospital Abnormal EEG Corpus (TUAB). PaSST achieved an accuracy of 95.7%, while LEAF an accuracy of 94.0%, Both significantly outperformed previously established benchmarks. The paper presents before the state of the art of the previous models for EEG Data and the cross-domain applications of audio techniques in EEG research. Original and modified models are described in order to understand how the transfer is possible from the audio to the EEG domain. This work provides evidence that machine learning accuracy can exceed inter-rater agreement levels between human experts.

**Weaknesses:**

Correct some typing and English errors
Just a tip to improve the reading of the article make a summary table of the two models before and after modifications with the main characteristics of each one.

---

### Decision · Program_Chairs · 2024-09-23

Accept